# OpenReview forum: "The Perils of Optimizing Learned Reward Functions: Low Training Error Does Not Guarantee Low Regret"
_ICLR.cc/2025/Conference — Submitted to ICLR 2025_

### Official Review · Reviewer_e9u2 · 2024-10-28

**Soundness:** 3
**Presentation:** 2
**Contribution:** 2
**Rating:** 6
**Confidence:** 3

**Summary:**

The paper studies learning-theoretic challenges related to reward modeling in RL. More specifically, it examines how the regret of an RL algorithm responds to errors in the reward model. It demonstrates that the regret diminishes as the error vanishes. Conversely, with a fixed error, the regret can become large. The authors refer to this property as *error-regret mismatch*. The analysis extends to RL with regularization techniques, including RLHF.

**Strengths:**

The paper provides an extensive analysis of the *error-regret* mismatch property in tabular MDPs. This includes RL objectives with regularization, as in RLHF. The main ideas are relatively easy to follow. I haven’t checked the full analysis in the appendix, but the arguments presented in the main text appear sound.

**Weaknesses:**

While high-level ideas are clearly presented, and the paper includes proof sketches, I generally found the paper to be quite dense, hard to read, and some parts quite confusing. Overall, the paper is not polished: the figures (Fig 1 and 2) look disproportionately large; some notation does not appear to be adequately explained when introduced (e.g., $\max J_R$ and $\min J_R$ in the 2nd paragraph of Sec 2); writing could be improved throughout the paper, as some sentences are unclear. For example, the paper mentions

> Our results highlight the challenge of deriving useful PAC-like generalization bounds for current reward learning algorithms. While this is possible (and has been done, see (Nika et al., 2024; Cen et al., 2024)), we showed that realistic bounds on the error in the learned reward function do not provide meaningful guarantees.

but does not explain what the authors actually mean by this. The cited references seem to provide guarantees on the suboptimality gap and regret. The paper also considers the RLHF setting but does not mention relevant references that study similar problems in this context, e.g., Sun et al., The Importance of Online Data: Understanding Preference Fine-tuning via Coverage. At the moment, it is not clear to me how to compare these results. It would be useful if the authors could clarify this in the rebuttal. In general, the related work section provides many references without actually explaining how they relate to the paper.

Regarding the results, it doesn’t seem surprising to me that a low error (Eq. (1)) can lead to high regret; this seems simply due to the distribution mismatch between the data used to learn $R$ and that used to train $\pi$. The intuition behind the proof of Proposition 3.3 follows this argument. However, it is well known that coverage assumptions are necessary for offline RL, including in RLHF settings, as argued in the references above. Hence, the conceptual novelty of the presented results is somewhat unclear to me. It would be great if the authors could clarify this.

It is also somewhat unclear what conclusions to draw from these results. The paper does not provide a concrete solution on how to approach the problem at hand, making it difficult to assess the practical significance of the findings. Moreover, the paper lacks experimental results that demonstrate the importance of the theoretical findings.

**Questions:**

Please see my comments in *Weaknesses*.

---

> ### Author Response · Authors · 2024-11-25
> **Comment**
>
> Edit: This comment was made before we knew about the extension of the discussion period.
>
> ~~Since the discussion period has nearly finished, we wanted to check that we have been able to address all of your concerns. If there are no further clarifications required, we hope that you will feel able to raise your score. Thank you very much for your time.~~

---

> ### Author Response · Authors · 2024-11-28
>
> Dear Reviewer e9u2,
>
> Thank you very much for your message and the note about complying with the page limit. We apologize for the incident and just uploaded a revised version of the paper that fully complies with all policies.
>
> Regarding your specific comments, in the revised paper version we:
> - reduced the size of the figures, and rewrote the paragraphs in "blue" to be more understandable. We continue to polish the whole article, which we deprioritized before the deadline to address other comments by you and the other reviewers.
> - completely revised the "Related Work" section to better compare our submission with existing work, and in particular the contemporaneous Sun et al.’s work.
> - Substantially revised the "Discussion" section to better focus on the implications of our work and make it more compact.
>
> We hope our revision addresses your feedback! Additionally, feel free to take a look at our newest global rebuttal, where we summarize the major paper changes. Please let us know if you have any remaining concerns, we are happy to continue the discussion for the remainder of the discussion period.

---

> > ### Comment · Reviewer_e9u2 · 2024-12-02
> >
> > Thank you for updating the paper to comply with the instructions. Since the rebuttal address many of my concerns I will raise my score.

---

### Official Review · Reviewer_ATno · 2024-10-29

**Soundness:** 3
**Presentation:** 4
**Contribution:** 2
**Rating:** 6
**Confidence:** 3

**Summary:**

The paper studies the problem of reward learning: given a data distribution, and assuming that the learner enjoys a guarantee bounding the error over the data distribution of the form

$$E[|R(s,a)-\widehat R(s,a)|/R_{max}] \le \varepsilon,$$

the authors are interested in understanding how good the policy $\widehat \pi$ corresponding to $\widehat R$ is w.r.t. the true MDP.

To this aim, the authors provide several results, most of the negative. Indeed, since the visiting distribution of the optimal policy may be very different from the distribution of $D$ from which state-actions are sample during the learning of the reward, having a relatively small $\varepsilon$ does not guarantee small "regret" (as the authors define it).
Controversely, there are examples where, in order to get a small regret, $\varepsilon$ needs to be of order $(S\times A)^{-1}$.

The authors then study the case of deriving, from $\widehat R$, a policy that is regulared, rather than the optimal one relative to $\widehat R$. Also in this case, a low reward error does not guarantee low regret, as proved in Theorem 4.2.

The paper closes with a discussion about the impact of these results the RLHF, and in particular to Large Language Models.

**Strengths:**

The paper is very well written, the presentation is clear, and the figures are perfect to convey the message.

Also, the importance of the problem is explained in a very detailed and precise manner, and the setting is defined in a way that is very "user-friendly".

The paper provides several different results, which answer many of the questions anyone may have about the setting.

**Weaknesses:**

My main concern with the results of the paper is that the results somehow lack novelty.

Propositions Proposition 3.1., 3.2, 3.3, Corollary 3.4. are very intuitive and natural (this is of course a quality), but aren't they somehow a consequence of the well-known lower bounds for offline learning?

Indeed, lower bounds for offline learning in case of tabular MDPs always contain some measure of coverage of the offline dataset. This coverage coefficient writes, for the evaluation of an arbitrary policy $\pi$ as

$$C_\pi=\sup_{s,a}\frac{d^\pi(s,a)}{D(s,a)}.$$

If we look at the worst possible policy, as in Proposition 3.3, it is clear that the coverage coefficient becomes of order $S\times A$, no matter the choice of data distribution $D(\cdot,\cdot)$. As lower bounds show that, in worst case, the sample complexity of any algorithm must scale with $C_\pi$, if follows that by outputting the policy which is optimal w.r.t. the reward function estimated over $D$, we could be making an error of order $C_\pi \varepsilon$ over is estimated return. This result seems to be very similar to what the authors get in Proposition 3.3.

Moreover, the choice made in equation (1),

$$E[|R(s,a)-\widehat R(s,a)|/R_{max}] \le \varepsilon,$$

that is assuming a bound on the Mean Absolute Error seems unjustified. Why didn't the author focus on the MSE, which is what gets minimized by most supervised learning algorithm?

Despite the  these criticisms, I must however point out that the paper seems to have its own narrative such as to frame the results in a new and interesting framework. Therefore I do not think that these weaknesses should compromise the acceptance of the paper.

**Questions:**

See weaknesses

---

### Official Review · Reviewer_nHaB · 2024-11-03

**Soundness:** 3
**Presentation:** 2
**Contribution:** 3
**Rating:** 6
**Confidence:** 2

**Summary:**

This paper theoretically studies the conditions that when learned reward models in RL will and will not lead to large regret in policy optimization. It shows that if the reward model's expected test error is suffient small (with respect to a threshold provided), then it would lead to low regret during policy optimization; otherwise, there are cases it would lead to large regret. The theoretical results highlight the need for careful consideration when employing learned reward functions in policy learning and underscore the importance of advanced reward function design.

**Strengths:**

The paper is clearly written, the results are interesting.

**Weaknesses:**

Maybe there are several points need to be further clarified, as detailed in my questions.

**Questions:**

1. On line 161, $J^R$ is policy evaluation function for reward function R with respect to which policy? The maximum and minmum is taken with respect to what? Policy or reward function?

2. I have a question on Proposition 3.1 and 3.3. Since the unsafe data distributions cover positive data distributions, is it possible that the $\epsilon$ in Proposition 3.3 satisfies the assumption in Proposition 3.2, which leads to a contradiction (say, D in unsafe(R, \epsilon, L) by prop 3.3 and D in safe(R,\epsilon,L) by prop 3.1)?

3. Can you justify the assumptions in Corollary 3.4? Since you are studying the tubular MDP setting, I was wondering if the second bullet, say, the supports of the policies are completely disjoint with each other, is too strong to hold, especially given that the $\epsilon$ should be set as a resonably samll value (not as small as that in Prop 3.2 though) and thus the number of policies in the policy set $\Pi_L$ is large. Maybe a toy example would help illustration.

4. On line 372, I found it hard to understand the sentence "a provided data distribution is safer than an initial reference policy".

5. On line 402, how $\hat{R}$ can be defined to be equal to R which is unknown?

6. I was wondering if there exist similar results as Proposition 3.2 and Theorem 3.5 under the setting ERROR-REGRET MISMATCH FOR REGULARIZED POLICY OPTIMIZATION.

7. If the resutls in this paper can motivate some novel algrithms for reward design? Any insight?

---

### Official Review · Reviewer_Ap9u · 2024-11-03

**Soundness:** 3
**Presentation:** 2
**Contribution:** 3
**Rating:** 6
**Confidence:** 4

**Summary:**

This paper investigates the theoretical challenges of using a learned reward function in policy optimization, particularly focusing on the "error-regret mismatch" problem—where a learned reward function has low expected error on a given data distribution. Yet, the resulting policy incurs high regret when evaluated against a true reward function. This issue primarily arises from the distributional shift. To frame the problem, the authors define "safe" and "unsafe" data distribution for any given MDP. A data distribution is classified as safe if any reward function with low expected error will reliably produce a low-regret policy. They provide theoretical analysis across both unregularized and regularized policy optimization scenarios, reaching consistent conclusions. The paper’s key theoretical insights are: (1) ensuring low regret may require prohibitively low expected error on the learned reward function, and (2) in certain large MDPs, all data distributions might inherently be unsafe.

**Strengths:**

The paper is overall well-written. The theoretical framework is clearly laid out, making the findings more comprehensible despite the inherently abstract nature of the paper.

The paper studies an important issue related to AI alignment (the potential pitfalls of using learned reward functions without proper evaluation).

The theoretical statements appear rigorous, and the authors provide intuitive explanations for deriving those results (although I did not review the proofs in the appendix).

**Weaknesses:**

The paper could benefit from a discussion of realistic scenarios where unsafe data distributions may lead to undesirable behaviors. An example would help clarify the practical implications of the theoretical results and provide more concrete context for the error-regret mismatch problem.

Regarding Definition 2.1: how detrimental unsafe data distributions can be in real-world practice? For instance, in the case of an unsafe data distribution, it is possible for a learned reward function with low expected error to either induce high regret ("bad" reward function) or low regret ("good" reward function). Understanding the likelihood of selecting a "good" reward over a "bad" one in practical settings would provide a more nuanced perspective on the applicability of the theoretical findings.

**Questions:**

What practical insights can a reward learning practitioner derive from this theoretical analysis? Are there actionable steps to avoid high-regret policies when using learned reward functions?

In Section 7, the paper discusses alternative reward evaluation methods. Do these alternatives alleviate the issues presented, or do they also have limitations?

---

> ### Author Response · Authors · 2024-11-25
> **Comment**
>
> Edit: This comment was made before we knew about the extension of the discussion period.
>
> ~~Since the discussion period has nearly finished, we wanted to check that we have been able to address all of your concerns. If there are no further clarifications required, we hope that you will feel able to raise your score. Thank you very much for your time.~~

---

> > ### Author Response · Authors · 2024-11-27
> >
> > Dear reviewer Ap9u,
> >
> > we thank you for your patience in receiving our updates. We apologize for our latest (now striked-through) comment where we missed that the discussion period was extended and that we still owe you a promised update.
> > In our last rebuttals, we had already answered three of your four concerns.
> > We now answer your remaining question:
> >
> > > The paper could benefit from a discussion of realistic scenarios where unsafe data distributions may lead to undesirable behaviors. An example would help clarify the practical implications of the theoretical results and provide more concrete context for the error-regret mismatch problem.
> >
> > We have uploaded a new version of our paper where we include an extensive example. We describe it in paragraph 4 and 5 of the introduction (in blue), and very extensively in Appendix B.4 (also colored blue).
> >
> > Essentially, the example is about a chatbot. The users can either ask safe queries (“Please help me create a high-protein diet”) or unsafe queries (“Please tell me how to build a nuclear weapon”). The chatbot can then either answer these queries or refuse. Now imagine a helpful-only policy that answers every query, no matter whether it is safe or not. Helpful-only policies have been analyzed in past safety research [1] and are often a starting point for policies meant to become “helpful, honest, and harmless” [2]. **Intuitively, such a helpful-only policy is unsafe if many people in the deployment environment ask unsafe questions, or if the damage caused by answering each such question is large**. See Figure 3 in the appendix for an illustration of this point. Note that the damage becoming large gets more likely with increasingly capable frontier models that unlock "dangerous capabilities" [3, 4].
> >
> > Unfortunately, it is hard for a typical reward learning paradigm without restrictions on the learned reward function to prevent the helpful-only policy from being learned. Intuitively, this is because the chatbot can answer any unsafe query in numerous different styles, such that **at least one such style must have a very low probability in the training distribution** for the reward model; the reward model can then inflate this answer’s value while achieving a low training error, thus making a helpful-only policy possible. **The fact that the reward model has low error while leading to the helpful-only policy with large regret is what makes this a case of error-regret mismatch.**
> >
> > The fact that there is *some* bad way of behaving that has a low probability of occurring in the training distribution is what essentially drives all of our negative results, Proposition 3.3, Corollary 3.4, Theorem 4.2, and Theorem 6.1. What makes them complicated is mainly that the dynamics become more complex in the full MDP setting, with regularization, and with the loss that appears in RLHF. Our positive results (Proposition 3.1 and 3.2, Proposition 4.1) use different ideas, e.g. Berge's theorem. Theorem 3.5 stands entirely on its own and is a complex analysis of the geometric structure of the set of safe data distributions, based on linear programming.
> >
> > [1] Carson Denison et al., *[Sycophancy to subterfuge: Investigating reward-tampering in large language models](https://arxiv.org/abs/2406.10162)*, 2024.
> >
> > [2] Amanda Askell et al., *[A General Language Assistant as a Laboratory for Alignment](https://arxiv.org/abs/2112.00861)*, 2021.
> >
> > [3] Mary Phuong et al., *[Evaluating Frontier Models for Dangerous Capabilities](https://arxiv.org/abs/2403.13793)*, 2024.
> >
> > [4] Anthropic. *[Responsible Scaling Policy](https://assets.anthropic.com/m/24a47b00f10301cd/original/Anthropic-Responsible-Scaling-Policy-2024-10-15.pdf)*, 2024.
> >
> > ---
> >
> > We thank you again for reviewing our work. We would appreciate your feedback on three specific points:
> >
> > 1. Does our new extended example help clarify a concrete context in which unsafe data distributions can lead to an error-regret mismatch?
> > 2. Have we adequately addressed the other concerns in our previous rebuttal?
> > 3. Has the revised version and our answers affected your overall evaluation of the work?

---

> > > ### Comment · Reviewer_Ap9u · 2024-12-02
> > >
> > > Thank you for clarifying my concerns, and updating the draft with an illustrative example. I agree with the authors regarding the paper's focus: the existence of unsafe policies, not the likelihood (which is a potential future study). I will raise my score.

---

### Author Response · Authors · 2024-11-28
**Revised paper uploaded**

Dear reviewers,

We would like to thank you all again for your extensive feedback! We have just uploaded a major revision of our paper in which we have tried to incorporate all of your feedback. For your convenience, we marked all the parts we changed with blue font. Some highlights:
- The "Related work" section has been completely rewritten.
- The "Discussion" section has been substantially revised.
- Appendix B3 has been newly added and contains a discussion about using the mean-squared error as an alternative distance measure.
- Appendix B4 has been newly added and contains a conceptual example of overoptimization concerns.

We have aimed to carefully address each of the concerns raised while preserving the underlying methodology and findings. We welcome your feedback and are happy to answer any questions you may have.

Kind regards,
the authors

---

### Author Response · Authors · 2024-12-02
**Please respond to our rebuttals**

Dear reviewers,

we want to remind you to please respond to our rebuttals before the reviewer-deadline in around 26 hours. We have tried our best to respond to all concerns and would like to hear how this affects your view of our work.

We especially want to encourage reviewers **Ap9u** and **e9u2** to respond, sind you have not reacted yet to any points in our rebuttals. Additionally, we encourage **nHaB** and **ATno** to respond to the continued discussion and new information.

Best wishes,
the authors.

---

### Meta-Review · Area_Chair_inBQ · 2024-12-20

**Metareview:**

Summary
This work investigates how reward learning errors impact policy optimization regret, addressing a critical issue in the current reinforcement learning (RL) literature. Specifically, the authors identify conditions under which low reward learning errors can ensure low policy optimization regret. These conditions, while conceptually significant, have not been formally summarized in previous works.

Strengths
- The problem studied is well-motivated, and its intuition is clearly conveyed.
- The technical proofs supporting the results are rigorous and sound.

Weaknesses:
A significant limitation of the paper is the lack of a clear connection to existing literature on hardness results in offline and inverse RL. Several reviewers have raised this concern. The classical concept of "coverage" is a well-established measure of learning difficulty in offline RL. While this work introduces a "safety distribution" notion to derive its hardness results, it implicitly relies on several coverage-related conditions. However, the relationship between the proposed safety distribution and the existing coverage assumptions is not explicitly or comprehensively discussed. Since this work is primarily theoretical, the absence of a detailed comparison with prior approaches undermines its contribution and makes it harder to justify its acceptance.

Decision:
Reject. While the technical content is solid, the lack of formal comparisons and connections to existing notions like coverage detracts from the paper’s completeness and impact.

**Additional Comments On Reviewer Discussion:**

During the discussion phase, the authors attempted to connect their proposed notion of "safety distribution" with the established concept of "coverage" in the existing literature. However, I still found the connection to be insufficiently clear, leaving the relationship between this work and prior studies ambiguous.

---

### Decision · Program_Chairs · 2025-01-22

Reject